# Influence of COVID-19 Crisis on Motivation and Hiking Intention of Gen Z in China: Perceived Risk and Coping Appraisal as Moderators

**DOI:** 10.3390/ijerph19084612

**Published:** 2022-04-11

**Authors:** Yunfan Wu, Keita Kinoshita, Yi Zhang, Rena Kagami, Shintaro Sato

**Affiliations:** 1Graduate School of Sport Sciences, Waseda University, 3-4-1 Higashifushimi Nishi-Tokyo, Tokyo 2020021, Japan; cloudfun@toki.waseda.jp (Y.W.); zhangyi@toki.waseda.jp (Y.Z.); rcy324.jump10@ruri.waseda.jp (R.K.); 2Faculty of Sport Sciences, Waseda University, 3-4-1 Higashifushimi Nishi-Tokyo, Tokyo 2020021, Japan; kkinoshita@aoni.waseda.jp

**Keywords:** leisure sport, push–pull motivation, perceived risk, coping appraisal, Generation Z, hiking, COVID-19

## Abstract

People’s lives have drastically changed since the outbreak of COVID-19. One concern during the pandemic has been the level of inactivity among people. Compared to various generations (e.g., baby boomers, generation alpha), Generation Z (Gen Z) traditionally spends much less time in outdoor spaces. Due to the pandemic, their inactiveness is assumed to be even more severe. Hiking, an outdoor activity, has become a possible remedy for young people to exercise in a safer sport environment compared to traditional facility-based activities. Although various studies have supported the link between motivations and hiking intention, the relationship may be altered based on psychological influences unique to the pandemic situations—perceived risk and coping appraisals. The current study was conducted to investigate the relationship between Gen Z’s motivations and hiking intention and moderating roles of perceived risk and coping appraisals in a pandemic environment. Data were collected from Gen Z between 18 and 24 in China (*N* = 407). The validity and reliability of all the constructs were assessed by confirmatory factor analysis (CFA), average variance extracted (AVE), and composite reliability. For testing hypotheses, PROCESS Macro 4.0 was used. The findings proposed that the appraisals of the pandemic situation (i.e., perceived risk and coping ability) moderated the relationship between two of the motivations—intellectual and destination motivations—and hiking intention. As a result, organizers of outdoor sports programs can implement viable strategies and take valid measurements to minimize the fear and worries among people in the time of the crisis.

## 1. Introduction

Since early 2020, the COVID-19 pandemic has spread fast globally and severely impacted people’s health [1]. One concern during the pandemic has been the level of inactivity among people, especially Gen Z. Gen Z, the “net” generation or the digital natives, is currently studying in secondary and higher education or has recently entered the employment market [2]. Segmenting the market by differentiating generational groups is a common way to understand consumer behavior better. The generation of baby boomers, Generation X, and Generation Y (Millennials) have been studied extensively. However, little attention has been paid to an emerging Gen Z. Gen Z has been treated as the world’s most influential consumer group, representing 40% of all consumers by 2020 [3]. According to the China statistical yearbook, Gen Z accounted for 10.74% in 2019 [4]. cCompared with other generations, Gen Z spends less time outside and easily loses a connection to nature, especially under the severe behavioral restrictions due to COVID-19. Louv [5] claimed that because these young people have been raised in a digital world, nature has gradually become something to watch (i.e., nature videos) or ignore. Declining nature participation may result in physical and mental issues. Since Gen Z is less vulnerable to COVID-19, they are recommended to spend time outdoors, in natural environments, and engaging in sport activities to maintain health and well-being in the short and long term [6].

A possible remedy that can help improve people’s inactivity in COVID-19 situations is outdoor sports due to a relatively safer environment than traditional facility-based activities. Data showed an increased rate of outdoor recreation participation, presumably due to the safer environment [7]. Among various nature recreation, day hiking has received keen attention [8]. As a nature-based outdoor activity, hiking is beneficial to mental health and well-being [9]. Wen et al. [10] also mentioned that hiking would allow participants to enjoy nature, contact others, and release the pressure derived from the pandemic in a relatively safer environment.

Before the outbreak of COVID-19 in China, hiking was a popular outdoor activity among Chinese citizens [11]. The Chinese government has adopted plenty of policies and measures fighting against the pandemic, encouraging people to return to everyday life based on the successful progress of COVID-19 control [1,12]. Wen et al. [10] claimed that outdoor recreation like hiking could serve as a traveling option for people during the holidays. Although various studies have supported the link between motivations and hiking intention, the relationship may be altered based on psychological influences unique to the pandemic situations—namely, perceived risk and coping appraisals. Under such an environment, does Gen Z still have the intention to go hiking? The literature does not include systematic and empirical investigations demonstrating how people’s perceived risk and coping appraisal influence their motivations and hiking intentions during the COVID-19 pandemic. Therefore, sports marketers must understand Gen Z’s consumption behaviors and what factors young consumers consider [13].

Motivation is an essential determinant of behavioral intention [14,15]. Prior researchers adopted the push–pull theory to interpret the influence of participants’ internal and external motivations on their attitude or behavioral intentions [15,16]. For example, a study conducted to understand Chinese people’s marathon participation found that internal motivation (i.e., excitement) was positively associated with their behavioral intention [17]. Sato et al. [18] also demonstrated that Japanese people’s behavioral intention (i.e., recommending adventure tourism destination) was positively associated with external motivations (i.e., cultural aspects of the destination and activity-related environment) in the context of white-water rafting. Happ et al. [19] showed that hikers’ external motivations (i.e., competition and exhibitionism) had significant associations with their attitudes toward hiking activities and intention to visit the destination. While internal motivations (i.e., social interaction and challenge) only influenced one’s attitude to the activities. Thus, there are mixed findings regarding the impact of push–pull motivations on behavioral intentions in tourism and leisure contexts.

We argue that the mixed findings regarding the relationship between motivations and behavioral intention are partly because of insufficient consideration of moderating variables. Due to the pandemic, individuals have started to pay more attention to risk and coping when making decisions about sport and physical activities. These constructs are imperative during unexpected risk events like the pandemic of COVID-19. Perceived risk appraisal refers to an individual’s perceptions of their susceptibility to harm. Previous studies found that perceived risk can play a moderating role in the relationship between motivations and behavioral intention in various contexts such as leisure and outdoor activities [20,21,22]. Rather, the authors of [23] also demonstrated that people’s risk perception can influence their behaviors. In addition, coping appraisal—the ability to cope with the potential loss or damage arising from the threat—is also an important cognitive component when unexpected crises occur [24]. The coping appraisal could moderate the relationship between motivation and behavioral intention [25,26,27,28]. Perceived risk and coping appraisal can interactively moderate the relationship between motivations and intention [29,30]. Nevertheless, such moderating roles have not been assessed in the crisis contexts like the COVID-19 situation. Therefore, this study aims to investigate the moderating roles of perceived risk and coping appraisal on the relationship between motivations and hiking intention in a pandemic environment by focusing on seemingly the most behaviorally restricted population—Gen Z.

### 1.1. Push–Pull Motivations and Hiking Intention

Sport participants’ motivations are multifaceted. Participation in recreational sports or physical activities includes a variety of motivations such as socialization, goal achievement, and escaping from boredom or daily life [31]. Prior scholars categorized motives into internal and external propositions [32]. Internal motivations are identified as internal psychological comprehensions that drive individuals to engage in certain behaviors. In contrast, external motivators are social or environmental factors that draw individuals to engage in certain behaviors [33]. For example, individuals are internally motivated when engaging in a particular activity driven by principles, feelings, and ambitions. Externally motivated individuals are driven by external environments such as advertising, media, and attributes of products and services.

The push–pull motivation framework has captured the multifaceted motivations in leisure and tourism. Push motivation, also called intrinsic motivation, is a fundamental and internal desire to get out from their living places to undertake travel [34]. Pull motivation, also called extrinsic motivation, is the external factors like destination attributes that attract people to visit the specific destinations over other places [34,35]. The push–pull motivation framework helps explain travel behaviors as to whether to go (push) and where to go (pull). Accordingly, the push–pull motivation framework can be applied to understand the socio-psychological decision-making process by internal desires and external forces [36,37].

The relationship between push–pull motivation and behavioral intention has been of significant interest in sport, events, and tourism [17,18]. For example, a previous study demonstrated that internal motivation, or excitement, positively correlates with intending to participate in, revisit, and recommend Chinese marathon events [17]. Khuong and Ha [38] also found that tourists’ returning intention to the destination is enhanced by the push and pull motivations. More consistent with the current study context (i.e., hiking), Sato et al. [18] found that push and pull motivations are significantly associated with sport tourists’ intentions to revisit the outdoor sport destination (i.e., white-water rafting destination).

**Hypothesis** **1a****:***Pull motivations are positively associated with hiking intention*.

**Hypothesis** **1b****:***Push motivations are positively associated with hiking intention*.

### 1.2. The Moderating Roles of Perceived Risk and Coping Appraisal

People worry, feel uncomfortable, and believe they are in danger due to the outbreak of COVID-19 [23,39]. In such situations, risk perception and coping appraisals are two important psychological factors that may influence behaviors. First, the authors of [23] defined the perceived risk as the perceived level of possible loss by an individual due to the global COVID-19 pandemic. As described previously in the literature on risk perception and health behavior, susceptibility plays a significant role in determining how individuals perceive risk [40]. Despite various cultures (e.g., ethnic background), regions (e.g., Eastern vs. Western), and countries [41], perceived risk has proven to be a vital predictor for health-protective behavior in many studies regarding respiratory infectious diseases [42]. Secondly, coping appraisal includes perceived self-efficacy (the belief that they can conduct the suggested behavior effectively to prevent the risk) and perceived response efficacy (the belief that they are performing the suggested action successfully to avoid the risk) [43].

Scholars have argued that perceived risk can moderate the relationship between motivations and behavioral intention [20]. Such moderating roles of perceived risk can be well explained based on the prospect theory [44]. Prospect theory suggests that individuals make decisions under risky and uncertain situations by evaluating potential gains and losses [45]. People tend to be more risk averse when the problem is framed as gains, whereas they tend to be less risk averse if they think they are at a loss [46]. In other words, people become more sensitive about losses compared to gains [47]. Prospect theory provides a comprehensive understanding of individuals’ decision-making process. Specifically, an individual’s evaluation of potential gains and losses will influence their preferences and behaviors under risk and uncertainty [29]. Different people evaluated the perceived gain and loss based on customers’ perceptions of specific behavior and individual differences [48]. As a result, people will have different reactions and psychological effects towards the same amount of loss [49].

Gains are associated with the actualization of tourism motivation, and losses are involved with risk [50]. Therefore, it is reasonable to consider gains and losses as push–pull motivation and perceived risk, respectively. People consider both motivations and risks when deciding whether or not to visit a focal destination [51], meaning that motivations and risk perceptions are interactively influencing behavioral intention. For example, in the context of upscale restaurants, motivations (i.e., intellectual and escape) significantly influence visit intention. This relationship is more prominent in consumers whose perceived risk is low [22]. Focusing on the Ebola case, Cahyanto et al. [52] also indicated that perceived risk influenced Americans’ avoidance of domestic travel behaviors. Similarly, Tavitiyaman and Qu [53] concluded that travelers’ risk perception of SARS and tsunami negatively influenced travel intention to Thailand. Based on the prospect theory and aforementioned empirical evidence, we posit the following:

**Hypothesis** **2a****:***The perceived risk moderates the relationship between push motivations and hiking intentions. The relationships are stronger for hikers who have low perceived risk than those who have high perceived risk*.

**Hypothesis** **2b****:***The perceived risk moderates the relationship between pull motivations and hiking intentions. The relationships are stronger for hikers who have low perceived risk than those who have high perceived risk*.

In addition to perceived risk, it is essential to consider the effect of coping appraisal. Coping appraisal refers to an individual’s assessment of his or her ability to cope with and avert the potential loss or damage arising from the threat [24]. For example, individuals can still decide to travel to the destination with risk if they can protect themselves by evaluating the situation and taking necessary protection actions. In the Protective Motivation Theory, the coping appraisal consists of (1) self-efficacy and (2) response efficacy. Self-efficacy is an individual’s perception of their capability to perform behaviors. Response efficacy refers to the perceived effectiveness of recommended risk preventative behaviors. Previous literature suggests that perceived self-efficacy and response efficacy can be associated with individuals’ behavioral intention. For example, tourists with high coping appraisal demonstrated stronger protection intention in the contexts of hospitality in Malaysia [54]. Focusing on food safety, Choi et al. [55] found that US college students with high perceived self-efficacy and perceived response efficacy tend to choose a safer restaurant to eat. These findings are in line with the food safety research of Crowley et al. [56], revealing that coping appraisal enhanced Americans’ intentions not to purchase irradiated food because of its potential health risks.

Prior research has shown that people with greater self-efficacy had stronger motivation to participate in physical activity [25,26]. Similarly, physical activity participants with high self-efficacy in the self-enhancing condition (i.e., intellectual motivation) do more exercise [28]. Therefore, based on what we mentioned above, the study puts forward the following:

**Hypothesis** **3a****:***The coping appraisal moderates the relationship between push motivations and hiking intentions. The relationship is stronger for hikers who have high coping appraisal than those who have low coping appraisal*.

**Hypothesis** **3b****:***The coping appraisal moderates the relationship between pull motivations and hiking intentions. The relationship is stronger for hikers who have high coping appraisal than those who have low coping appraisal*.

According to the Risk Perception Attitude (RPA) framework [40], those who have solid coping appraisal are likely to treat potential risks as challenges to be overcome, whereas those lacking coping appraisal typically think they have a great possibility to risk [57]. Four groups are formed based on individuals’ risk perceptions and coping appraisal [29]. They are (1) the responsive attitude group (high perceived risk, high coping appraisal), (2) the avoidance attitude group (high risk perceptions, low coping appraisal), (3) the proactive attitude group (low risk perceptions, high coping appraisal), and (4) the indifference attitude group (low perceived risk, low coping appraisal), suggesting that risk perception and coping appraisal are mutually interactive. Researchers found that if the environment gave a specific level of risk, those with more robust coping appraisal were likely to show more positive health intention than those with lower coping appraisal [30]. In the research of food delivery under the COVID-19, Leung and Cai [28] revealed that coping significantly moderated the relationship between perceived risk and purchase intention. Specifically, higher self-efficacy customers are more likely to order digital food deliveries even facing perceived risk. When it comes to watching videos, Wong and Yang [58] found a significant moderating effect of self-efficacy on the relationship between watching positive risk-taking videos and risk-taking intention. Individuals with higher self-efficacy will have lower risky behavior when perceiving higher risk. Wang et al. [59] also indicated a significant moderating effect of self-efficacy on the relationship between perceived value and purchase intention. In the current study, the perceived risk captures one’s expectation of exposure to the COVID-19 virus during hiking. For instance, when someone goes hiking and feels serious about the threat to health, the individual has a high likelihood of feeling risky. The person may respond to this situation and take measures to reduce the threat effectively. Some of them will be confident in their abilities to deal with such a situation. Therefore, they will still want to go hiking even with the COVID-19 threat to their health. However, some may not be confident of their ability, so their intention for hiking will be influenced. We hypothesize that the effect of perceived risk on people’s push–pull motivations and hiking intention would be moderated by their coping appraisal.

For these reasons, we tested the propositions of the RPA framework in the context of push–pull motivation and hiking intention. To explain the interaction between these variables, the role of coping appraisal as a moderator is also analyzed. The study tried to determine the relationship between two independent variables (push and pull motivations) and a dependent variable (hiking intention) with two moderating variables (perceived risk and coping appraisal).

**Hypothesis** **4a****:**
*Perceived risk and coping appraisal will interactively moderate the relationship between push motivations and hiking intention. Specifically, the relationship between push motivations and hiking intention among individuals with low (high) perceived risk will be stronger (weaker) when coping appraisal is high (low).*


**Hypothesis** **4b****:**
*Perceived risk and coping appraisal will interactively moderate the relationship between pull motivations and hiking intention. Specifically, the relationship between pull motivations and hiking intention among individuals with low (high) perceived risk will be stronger (weaker) when coping appraisal is high (low).*


To explore predictors and their moderation relationship and influence on hiking intention for Gen Z, we integrated critical concepts from the push–pull motivation theory, prospect theory, RPA theory, and additional variables from the empirical literature. In addition, the study used a moderated moderation model to check the effect. The conceptual model can be found in Figure 1. Combining the prospect theory, push–pull theory, and RPA theory, the current study investigates the relationships between Gen Z’s motivation and hiking intentions and assesses the moderating roles of perceived risk and coping appraisal.

## 2. Materials and Methods

### 2.1. Research Design, Sampling Plan, and Data Collection

An online survey was employed to obtain data from residents in China. This study examines participants of Gen Z, which refers to individuals aged between 18 and 24 years old (age during collection time in 2021). Gen Z, the first generation born to be digital natives, is now entering college and the workforce [60]. As a result, they are accustomed to technologies and good at using technologies better than their former generations [61]. Furthermore, Gen Z is rapidly growing as a source of visitors to various tourism sites, and they are predicted to become the essential tourism consumption group in the future [62]. Therefore, understanding the preferences of Gen Z is vital to gain an advantage in developing sports marketing plans to attract the target group.

We recruit respondents on a popular Chinese online survey platform (Tencent Survey Platform). Participants received a small monetary compensation for their voluntary participation. The questionnaire comprises of two sections. The first section contains push and pull motivations (intellectual, social, mastery, avoidance, and destination attributes), the scale of perceived risk, perceived coping appraisal (perceived self-efficacy and perceived response efficacy), and hiking intentions. The second section contains personal profile questions (e.g., age, gender, and educational level). Before starting the questionnaire, participants were provided with written information, which described the research purposes, confirmed that all responses would be confidential, and explained that participants had the right to withdraw from the study.

We initially collected 604 questionnaires from participants with age restrictions that fell under the definition of Gen Z. We then excluded participants who spent less than two minutes answering the questionnaire based on the assumption that they were inattentive. After further excluding participants who had never participated in hiking at data collection, we obtained 437 responses. After further screening out 30 outliers by Mahalanobis distance [63], cook’s [64], and leverage values [65], 407 responses were analyzed at the end (male = 29%, female = 71%; *M_age_* = 20.69, *SD_age_* = 1.70; college students = 73%). The sample size of 407 is sufficient in measuring the complex relationships of the proposed model. With 407 totaling the usable sample size and 31 indicators of variables, the ratio of the number of cases to the number of free parameters was 15:1. This acceptable ratio could indicate the statistical precision of the results [66].

### 2.2. Measurements

Regarding push motivations, we measured four factors (i.e., intellectual, social, mastery, and avoidance) adopted from the previous literature [18,67]. Four items measured intellectual motivation (e.g., *to explore new ideas*) [68,69]. Two items captured social motivation (e.g., *building friendships and interacting with others*) [18,67]. We used three items to measure mastery motivation (e.g., *to challenge my abilities*) [70]. Two items measured avoidance motivation (e.g., *to relieve stress and tension*) [69]. For the pull motivation, the study adopted one factor (destination) with five items to measure (e.g., *the wilderness and nature of the hiking destination*) [71]. All items regarding push–pull motivations were measured on a 7-point Likert-type scale ranging from 1 “extremely unimportant” to 7 “extremely important.”

As for perceived risk, we used three items adapted from the previous literature (e.g., *I am at risk of being a victim of COVID-19 while hiking*) [72,73]. Participants answered the questions based on a 7-point Likert-type scale ranging from 1 “extremely unlikely” to 7 “extremely likely.” Coping appraisal consisted of perceived self-efficacy and perceived response efficacy. Four items were used for perceived self-efficacy, adopted from previous literature (e.g., *How confident are you to perform the following actions to ensure your personal safety while hiking?*) [72]. Perceived self-efficacy was measured on a 7-point Likert-type scale from 1 “not confident at all” to 7 “very confident” with four items. Four items were used for perceived response efficacy, adopted from previous literature (e.g., *preparing anti-COVID equipment and coping measures*) [72]. Perceived response efficacy was measured on a 7-point Likert-type scale from 1 “not effective at all” to 7 “very effective” with four items.

Hiking intention was measured with a three-item scale, adapted from vacation intention research (e.g., *How likely is it that you will go hiking during the current situation?*) [74]. All items were evaluated on a 7-point scale, ranging from 1 “not willing at all” to 7 “very willing” with three items. It is also important to note that bilingual researchers used back-translation to assess all items mentioned above as they were initially developed in English.

### 2.3. Data Analysis

Descriptive statistics were generated in SPSS 27.0 (IBM, Armonk, NY, USA). Confirmatory factor analysis (CFA) was carried out using Amos 27.0 (IBM, Armonk, NY, USA). To test our hypothesis, we used conditional process modeling (PROCESS macro) for SPSS. PROCESS macro enabled us to conduct moderated moderation (Hayes Model 3) tests to assess the indirect effects of push–pull motivations on hiking intention by moderating perceived risk and coping appraisal.

CFA was applied to estimate the appropriateness of the measurement model fit. Based on the previous literature [75], the study employed ordinary goodness of fit indices, including the comparative fit index (CFI), the Tucker–Lewis coefficient (TLI), the root-mean-square error of approximation (RMSEA), and the standardized root-mean-square residual (SRMR) to measure the estimation model fit. We also used factor loadings and the average variance extracted (AVE) to assess convergent validity [76]. Discriminant validity was also tested by comparing each construct’s AVE and squared correlations.

PROCESS macro was used to test the hypothesized relationships involving moderators [77]. Education and gender were included as demographic covariates to control participants’ characteristics. Education was coded as junior school and below (1), high school/technical school (2), college degree (3), Bachelor’s degree (4), and Master’s degree and above (5). Gender was coded as male (0) and female (1). These two covariates were examined as possible confounders of the effects on hiking intention. The principle of this algorithm is based on multiple linear regression and is considered the most appropriate for analyzing interactions between one or more independent variables [78]. For example, two moderators interact using a series of Model 3 with 95% confidence intervals and 5000 bootstraps [77]. Thereby, all hypotheses could be analyzed simultaneously.

## 3. Results

### 3.1. Measurement Model Assessment

The results of descriptive statistics are presented in Table 1 and Table 2. We conducted confirmatory factor analysis (CFA) to assess the measurement model. The results in Table 2 demonstrated that the data fit well with the model (*χ2/df* = 1185.56/393 = 3.02; *CFI* = 0.90; *TLI* = 0.88; *RMSEA* = 0.07; and *SRMR* = 0.06) [75,79]. Composite reliability (CR) values were calculated for assessing internal consistency. All CR values were above 0.70, indicating that the internal consistency was acceptable [80,81].

Factoring loadings and average variance extracted (AVE) were calculated to test convergent validity. If factor loadings are high and converge on a common point, the constructs are considered to share a high convergent validity [82]. All standardized factor loadings ranged from 0.62 to 0.89 and were significant at the *p* < 0.001 level, suggesting that convergent validity was deemed acceptable. Each construct’s AVE ranged from 0.46 to 0.71. All AVE values, except mastery and destination motivation, were greater than 0.50 [76]. Although mastery and destination motivation indicated slightly less than the threshold, we considered that convergent validity was ensured overall [83].

Discriminant validity was evaluated by comparing the square of the correlations and AVEs. As shown in Table 3, most values of the AVEs (on the diagonal) are greater than their respective correlation estimates. This comparison provides evidence that discriminant validity was ensured [76]. 

### 3.2. Hypotheses Testing

We examined the direct relationships between push–pull motivations and hiking intention. Meanwhile, we assessed the moderating roles of perceived risk and coping appraisal and the moderated moderation effects of the two variables on the relationship between push (i.e., intellectual, social, mastery, and avoidance) and pull motivations (i.e., destination) and hiking intention.

Hypotheses 1a,b posited positive associations between push–pull motivation and hiking intention. There was non-significant simple positive effect of push motivations on hiking intention. It was revealed that push motivations (i.e., intellectual, social, mastery, and avoidance) did not have any directly significant relationship with hiking intention (*B* = −1.45, *t* = −1.72, *p* =0.09, *CI* = [−3.10, 0.20]; *B* = −0.53, *t* = −0.61, *p* = 0.55, *CI* = [−2.27, 1.20]; *B* = −0.54, *t* = −0.59, *p* = 0.56, *CI* = [−2.36, 1.27]; *B* = −0.60, *t* = −0.67, *p* = 0.51, *CI* = [−2.36, 1.17], respectively). The effect of pull (i.e., destination) motivation on hiking intention (*B* = −1.79, *SE* = 1.00, *t* = −1.80, *p* = 0.07, *CI* = [−3.73, 0.14]) was not significant either. Therefore, Hypotheses 1a,b were not supported. 

As for Hypotheses 2a,b, the interaction between perceived risk and push (intellectual) motivation on hiking intention was significant (*B* = 0.49, *p* = 0.03); the interaction between perceived risk and pull (destination) motivation on hiking intention was also significant (*B* = 0.61, *p* = 0.03). However, the other push motivations (i.e., social, mastery, and avoidance) had no significant two-way interaction. The relationship between push (intellectual)/pull (destination) motivations and hiking intention was stronger for those who perceived higher risk, rejecting Hypotheses 2a,b.

Hypotheses 3a,b assumed that the coping appraisal could moderate the relationship between push–pull motivation and hiking intention. The interaction between coping appraisal and push (intellectual) motivation on hiking intention was significant (*B* = 0.32, *p* = 0.02); the interaction between coping appraisal and pull (destination) motivation on hiking intention was also significant (*B* = 0.37, *p* = 0.02). The relationship between push (intellectual)/pull (destination) motivations and hiking intention was stronger for those who perceived higher coping appraisal. Likewise, the other push motivations (i.e., social, mastery, and avoidance) had no significant two-way interaction. Therefore, Hypothesis 3a was partially supported, and Hypothesis 3b was supported.

Hypotheses 4a,b assumed that the coping appraisal components (perceived self-efficacy, perceived response efficacy) influenced the moderating effect of perceived risk on the push–pull motivation and hiking intention relationship. The results revealed that the three-way interaction involving push motivations (i.e., social, mastery, avoidance), perceived risk, and coping appraisal was not significant for hiking intention (*B* = −0.04, *p* = 0.28; *B* = −0.03, *p* = 0.39; *B* = −0.03, *p* = 0.50). Nevertheless, the three-way interaction among one of the subdimensions of push motivation (i.e., intellectual), perceived risk, and coping appraisal on hiking intention was significant (*B* = −0.08, *p* = 0.02). Similar results showed that the three-way interaction of pull motivation (i.e., destination), perceived risk, and coping appraisal on hiking was significant (*B* = −0.10, *p* = 0.02). Consequently, Hypothesis 4a was partially supported, and Hypothesis 4b was supported. More details about the examinations of moderated moderation are reported below.

### 3.3. Moderated Moderation for Push Motivation and Hiking Intention

We examined whether the coping appraisal influenced the moderating effect of perceived risk on push motivation and hiking intention relationship. As shown in Table 4, the overall model of intellectual motivation was significant, which explained 34.5% of the variance in hiking intention (*R*^2^ = 0.37, *MSE* = 0.67, *F*(9, 397) = 25.42, *p* < 0.001). A significant three-way interaction effect was observed (Intellectual motivation × Perceived risk × Coping appraisal) for hiking intention. It means that the moderation of push (intellectual) motivation on hiking intention by perceived risk was a function of the coping appraisal (*B* = −0.08, *t* = −2. 29, *p* = 0.02).

The Johnson–Neyman method indicated that the interaction between perceived risk and coping appraisal moderated the push (intellectual) motivation and hiking intention over part of the range of coping appraisal scores. There was a significant positive two-way interaction between push (intellectual) motivation and perceived risk when scores of coping appraisal were lower than 3.11 (*B* = 0.24, *95% CI* = 0.00, 0.48, *p* = 0.05) or higher than 6.86 (*B* = −0.06, *95% CI* = −0.13, 0.00, *p* = 0.05). The effect of push (intellectual) motivation on hiking intention was consistently positive. However, the difference in its effect between low perceived risk and high perceived risk was slightly larger among those with a weak coping appraisal. The moderation of the interaction between intellectual motivation and perceived risk by coping appraisal only accounted for 0.8% of the variance in hiking intention.

The conditional positive effect of motivation on hiking intention as a function of perceived risk and coping appraisal is presented graphically in Figure 2 with three panels corresponding to values on coping appraisal equal to 16th, 50th, and 84th percentiles. At low levels of coping appraisal, the low perceived risk potentiated the effect of intellectual motivation on hiking intention, whereas at high levels of coping appraisal, the intellectual motivation–hiking intention relationship buffered for high but not low perceived risk. Specifically, the relationship between push (intellectual) motivation and hiking intention with low perceived risk was stronger when coping appraisal is high; the relationship between push (intellectual) motivation and hiking intention among individuals with high perceived risk was stronger when coping appraisal is low.

As shown in Table 4 in regard to the rest push motivations—social, mastery, and avoidance motivation—no significant main effects or two- or three-way interactions were observed. The covariate, education, was non-significant for hiking intention, including for intellectual motivation (*B* = 0.02, *t* = 0.30, *p* =0.76, *CI* = [−0.08, 0.11]), for social motivation (*B* = −0.02, *t* = −0.28, *p* = 0.78, *CI* = [−0.13, 0.09]), for mastery motivation (*B* =0.03, *t* =0.54, *p* = 0.59, *CI* = [−0.07, 0.12]), and for avoidance motivation (*B* < 0.01, *t* = 0.03, *p* = 0.97, *CI* = [−0.10, 0.11]). The lack of an interaction for education suggests that despite different education background, the psychological process of perceived risk and coping appraisal works in a similar way on intention across Gen Z.

The covariate, gender, was significant for hiking intention, including for intellectual motivation (*B* = −0.33, *t* = −3.56, *p* < 0.01, *CI* = [−0.51, −0.15]), for social motivation (*B* = −0.38, *t* = −3.68, *p* < 0.01, *CI* = [−0.58, −0.18]), for mastery motivation (*B* = −0.31, *t* = −3.51, *p* < 0.01, *CI* = [−0.49, −0.14]), and for avoidance motivation (*B* = −0.38, *t* = −3.85, *p* < 0.01, *CI* = [−0.57, −0.18]). Gender negatively affected hiking intention. Specifically, male participants have more possibility to go hiking compared to female regardless of any kinds of motivations.

### 3.4. Moderated Moderation for Pull (Destination) Motivation and Hiking Intention

We examined whether the coping appraisal components (perceived self-efficacy, perceived response efficacy) influenced the moderating effect of perceived risk on the pull motivation–hiking intention relationship. As Table 5 shows, the overall model of pull (destination) motivation was significant (*R*^2^ = 0.31, *MSE* = 0.73, *F*(9, 397) = 19.37, *p* < 0.001). PROCESS 4.0 yielded a three-way interaction (Destination motivation × Perceived risk × Coping appraisal) for hiking intention (*B* = −0.10, *p* = 0.02). The Johnson–Neyman method indicated that the destination by perceived risk (two-way interaction) was significant when the coping appraisal scores were lower than 3.94 (*B* = 0.22, *95% CI* = 0.00, 0.45, *p* = 0.05). The effect of pull (destination) motivation on hiking intention is consistently positive. However, the difference in its effect between low perceived risk and high perceived risk is larger among those with a weak coping appraisal. The moderation of the interaction between motivation and perceived risk by coping appraisal only accounts for 0.9% of the variance in hiking intention.

The conditional positive effect of motivation on hiking intention as a function of perceived risk and coping appraisal is presented graphically in Figure 3 with three panels corresponding to values on coping appraisal equal to 16th, 50th, and 84th percentiles. At low levels of coping appraisal, perceived risk potentiated the effect of pull (destination) motivation on hiking intention, whereas at high levels of coping appraisal, the pull (destination)–hiking intention relationship was buffered for high but not low perceived risk. Specifically, the relationship between pull (destination) motivation and hiking intention among individuals with high perceived risk was stronger when coping appraisal is low; the relationship between pull (destination) motivation and hiking intention among individuals with low perceived risk was stronger when coping appraisal is high.

The covariate, education, was non-significant for hiking intention (*B_destination_* < 0.01, *t* = 0.08, *p* = 0.94, *CI* = [−0.10, 0.11]). The lack of an interaction for education suggests that despite different education background, the psychological process of perceived risk and coping appraisal works in a similar way on intention across Gen Z.

The covariate, gender, was significant for hiking intention (*B_destination_* = −0.34, *t* = −3.52, *p <* 0.01, *CI* = [−0.53, −0.15]). Gender negatively affected hiking intention. Specifically, male participants have more possibility to go hiking compared to female regardless of any kinds of motivations.

## 4. Discussion

This study aimed to examine relationships between push–pull motivation and hiking intention and moderation effects of perceived risk and coping appraisal under the context of the COVID-19 situation. The push–pull theory acted as a theoretical guide to serve this purpose. Furthermore, the study tested the moderated moderation model for perceived risk and coping appraisal by analyzing a survey conducted with Gen Z. The study found some potential moderating effects and practical implications for sport organizers of hiking destinations in China. 

First, results indicated that none of the motivations is associated with hiking intention. Push and pull motivation had no significant relationship with hiking intention. In other words, young people’s needs for mental activities, social desire, the pressure from daily life, and wishes to develop their ability cannot become a powerful reason to inspire young people’s positive intention to go hiking. Additionally, the attraction of hiking destinations in the COVID-19 pandemic did not stimulate young people’s positive intention to go hiking. The findings are inconsistent with Khuong and Ha [38], and Sato et al. [18], who found that push and pull motivation positively predicted intention. As we mentioned, Gen Z is known for being digital natives and the “net” generation. Many motivations in this research can be satisfied at home by technologies instead of going outside to suffer danger. Given the harsh reality, push–pull motivations may not be enough to get them to go hiking. For example, people can brainstorm online or surf the internet to get new ideas; video calls and network applications can help they meet their social needs; they can attend online lessons or circles activities to work out under the guidance of private online coaches; watching movies, reading books, playing games, or meditation can help people get rid of stress and pressure. During the pandemic, quarantine rules, traffic inaccessibility, and other restrictions stimulated negative and resistant feelings about hiking. People can experience online VR trips to destinations instead of visiting the place in person. Thus, push–pull motivation did not associate directly with hiking intention during the pandemic.

Second, the results confirmed that the perceived risk positively enhanced the relationship between push (intellectual) and pull (destination) motivations and hiking intentions. The higher the perceived risk, the stronger the relationship between push (intellectual) and pull (destination) motivations and hiking intentions. People who are perceived as high risk are more willing to go hiking than those with low perceived risk. The result is not in line with the previous research [22], showing that the relationship between motivations and visit intention can be enhanced if the perceived risk level is low. Those who perceived higher risk have a stronger intention to go hiking. Prospect theory can provide some explanation of the decision-making process that potential gains (motivations) and losses (risk) will be assessed under risk and uncertainty [29,44]. Due to various perceptions and differences, people appraise the gain and loss in different results [48]. Thus, even with the same amount of loss (risk), people would react differently and have various psychological activities [49]. Gen Z is not a vulnerable group. They may think “high” risk is not a big deal based on personal hiking experience, so they still want to go hiking.

Third, the coping appraisal moderated the relationship between push (intellectual) and pull (destination) motivations and hiking intentions. The higher the coping appraisal, the stronger the relationship between push (intellectual) and pull (destination) motivations and hiking intentions. The coping appraisal positively affected the relationship, confirming Annesi’s [25] finding that self-efficacy has enhanced the relationship between exercise motivation and intention of physical activity. Combined with the educational campaign or propagation from schools and society, they know that they are not a vulnerable group and learn how to protect themselves correctly. Strong coping appraisal gave them the confidence to believe that they can protect themselves sufficiently and avoid getting the disease. That is why the relationship becomes stronger with the increasing of coping appraisal.

Fourthly, perceived risk and coping appraisal interactively moderated the relationship between push (intellectual) and pull (destination) motivations and hiking intention. Specifically, the perceived risk only moderated the relationship between push (intellectual) motivation and hiking intention when the coping appraisal was higher than a certain score (*M = 6.87*) or lower than a certain score (*M = 3.25*); additionally, the risk only moderated the relationship between pull (destination) motivation and hiking intention when the coping appraisal was lower than a certain score (*M = 4.13*). People with a high coping appraisal score who perceived low risk would have a higher intention of hiking than those who perceived high risk. While in the low coping appraisal, people who perceived high risk would also have stronger intentions than those who perceived low risk. The results fit the Risk Perception Attitude (RPA) framework that individuals can be divided into groups according to their level of risk perceptions and coping appraisal [30]. Responders can be divided into four segments based on their perceived risk and coping appraisal. Responsive people (high risk, high copping appraisal) are most likely to adopt self-protective behaviors. Proactive people (low risk, high copping appraisal) occasionally tend towards self-protective behavior. Avoidant people (high risk, low copping appraisal) experience conflicted feelings, making them less likely to take measures to protect themselves. Finally, indifferent people (low risk, low copping appraisal) generally take minor protective measures.

Gen Z is born in the digital era and is good at using technology tools to search and learn information, which means their information channels are broad and accessible. Although it is hard to guarantee the quality of information, they can get the latest news faster and know emergencies first. They may react to this situation and effectively reduce the threat. In the meantime, solid coping appraisals gave Gen Z the confidence to overcome difficulties during hiking. They believed themselves and the measurements or equipment they used were effective. All these factors made them believe they are less likely to catch the virus, and they will consider all the aspects when they make judgments. When people have a high coping appraisal, perceived risk will influence push (intellectual) motivation and hiking intention. In this situation, the low perceived risk group with high coping appraisal was more willing to go hiking in such a pandemic environment. When people have a low coping appraisal, perceived risk will also influence the push (intellectual) motivation and pull (destination) motivation on hiking intention. In the traditional wisdom, hikers with a lower risk of getting the disease tend to be more willing to go hiking. However, we found it to be adverse: people with a higher risk appraisal of the pandemic wished to attend hiking activities despite their low coping appraisal towards the COVID-19 pandemic.

The framework RPA indicated that the avoidant group (high risk, low copping appraisal) was likely to take measures to protect themselves. However, indifference groups (low risk, low copping appraisal) were unlikely to take protective measurements. Responders in the avoidant group may take steps to protect themselves from having a higher intention to go hiking. Furthermore, some people may be overconfident and overestimate their own ability to deal with such a situation; others may lack enough recognition of pandemics. That might be why some people still want to go hiking despite the COVID-19 threat to their health. From these results, we can see that coping appraisal plays an essential role in the study. A person with an excellent coping appraisal is likely to look at potential risks as challenges to be overcome. In contrast, those with an inadequate coping appraisal are likely to think they have great potential for risk [22,57].

One thing needs to be mentioned, a demographic factor like education does not influence people’s intention. It means that the decision-making process is similar among Gen Z despite the different education background of Gen Z individuals. However, it is different when males and females are making decisions. Males are prone to have high intention to go hiking compared with females under the same situation.

### 4.1. Implications

Some practical implications can help hiking destination organizers reduce worries about risk to attract residents and outsiders to engage in outdoor activities. First, the results of this study demonstrated that coping appraisal is essential for increasing hiking intention. From a management perspective, organizers of hiking destinations should consider establishing a healthy and hygienic environment to reduce people’s worries, influencing how hikers evaluate the destinations. The COVID-19 situation stipulates that people can choose the destination that best suits their needs and decide whether the trip duration, distance to the destination, or the mode of transportation should be for them. When making decisions, people will be highly concerned with hygiene and physical health [84], disinfection, and a reliable health system in a destination [85]. The destination can provide detailed information for visitors before traveling and recommend visitors to purchase travel insurance and receive vaccines [86]. Previous research showed that people’s past experiences could become a potential driving factor to stimulate future consumption [87]. Providing better sport services based on consumers’ benefits can promote sport consumers’ psychological experience and well-being and then build enduring relationships to encourage re-consumption [88].

Besides, it is essential to use digital technology and the internet to promote destinations and attract Gen Z. These young internet natives have less associations with nature, but they still have a strong intention to do outdoor activities. Based on different needs of gender, destinations can design various routes to satisfy people’s requirements. They can dig out the features of places and stimulate the interest of potential visitors. It is important to target and categorize the different groups. For the segment of high-risk groups, society should provide them with education campaigns about preventive measures and make sure these young people are aware of the risks of COVID-19. It is essential for other groups to get accurate, current information to encourage them to take action. They do not need a motivation to act but need information about what to do and how to do it safely [73].

Furthermore, hiking has substantial economic benefits and social influences that are hidden. Hiking can generate public participation with many economic and social benefits, despite its relative lack of competitiveness and attraction compared to professional sport [89]. Hiking may attract many participants, promoting sports tourism development in destinations and encouraging enterprises to sponsor excursions in rural areas. Rural areas and organizers of destinations should be aware of the potential economic impact of hiking and treat it as a countryside development strategy. At the same time, outdoor activities should be marketed to promote public health by attracting women, older adults, and overweight people. Immersing in nature is good for both mental and physical health. Hiking is a recreational activity characterized by a non-competitive nature; thus, it does not necessarily require intensive training for individuals and costs to be organized. It usually happens in open, uncrowded, and airy rural areas. This outdoor activity can be better than going to the gym during the pandemic. Organizers of hiking destinations should note that the health benefits of hiking can act as a catalyst to promote physical activity. Hikers who gain better leisure and entertainment experiences will become more likely to return. Hence, making great efforts to promote hiking or other outdoor activities is one way to build a healthy China.

Overall, the study offers better implications for understanding the impact of COVID-19 on hikers’ intentions. Marketers should understand how to deal with this new challenge and adapt to the latest situation. In addition, the current study can also contribute to sports marketing and management by providing valuable suggestions for organizing hiking activities in the future. Sports marketers and sports tourism managers should consider marketing strategies to build a healthy and safe hiking environment.

### 4.2. Limitations and Future Research Directions

There are some limitations to the study. First, although we focused on Gen Z in China, respondents in the online panel we used for this study demonstrated a significant gender imbalance. In this sense, researchers should exercise caution regarding the generalizability of the findings. In addition, hiking is a natural activity that everyone can participate in easily. Future research should examine other generations, including the elderly, to accumulate more generalizable evidence.

Second, our conceptual model was limited in its scope of risk perception and coping appraisals. Regarding the relationship between motivations and behavioral intention in nature-based activities, future research should consider various factors that can mediate or/and moderate the relationship. In this sense, considering individuals’ psychological factors such as vitality and learning [90], religious or spiritual capital [91], and mental toughness [92] could be an exciting research endeavor, particularly in the restricted situations.

Third, the current research employed a cross-sectional design. Though it is somewhat cliché, as causal relationships cannot be established. Future research that employs longitudinal data would greatly contribute to the literature. Since the outbreak has already occurred, it is impossible to obtain Gen Z’s pre-pandemic behavioral tendencies. Nevertheless, scholars can deepen their knowledge regarding their hiking participation behaviors by comparing them during the pandemic and post-outbreak periods.

Lastly, the current research focused on behavioral intention only. Considering the challenging situation that people experience, one of the most important dependent variables can be well-being. Sport and leisure activities, particularly in outdoor spaces, are essential in boosting people’s well-being [93,94]. Future studies should be conducted to understand further the effects of nature-based activities, which would be helpful information for policymakers. 

## 5. Conclusions

Since lockdown and social distancing have affected the selection of outdoor activities, people’s behaviors have changed. The COVID-19 pandemic has had an enormous impact on the leisure industry. Drawing upon push–pull theory, prospect theory, and the RPA framework, we examined that the moderation effect of perceived risk and coping appraisal on the relationship between push–pull motivation and hiking intention, which can vary substantially in Gen Z. Perceived risk and coping appraisal can interactively moderate the push–pull motivation–hiking intention link. Perceived risk was only moderated when the coping appraisal is in specific range. Thus, this research stressed the coping appraisal for the importance of practice management. Although Gen Z is less connected with the nature, they still wish to attend hiking activities during the COVID-19 pandemic. Outdoor sport organizers should communicate with Gen Z by different channels to promote their destinations and recommend young people to come out in the nature to maintain their health and well-being.

## Figures and Tables

**Figure 1 ijerph-19-04612-f001:**
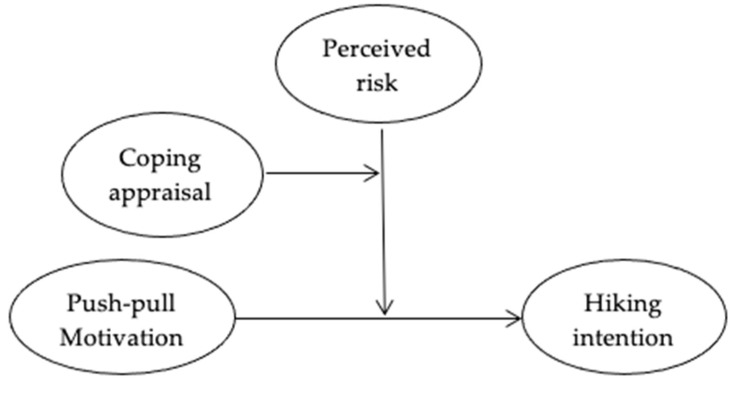
Conceptual framework.

**Figure 2 ijerph-19-04612-f002:**
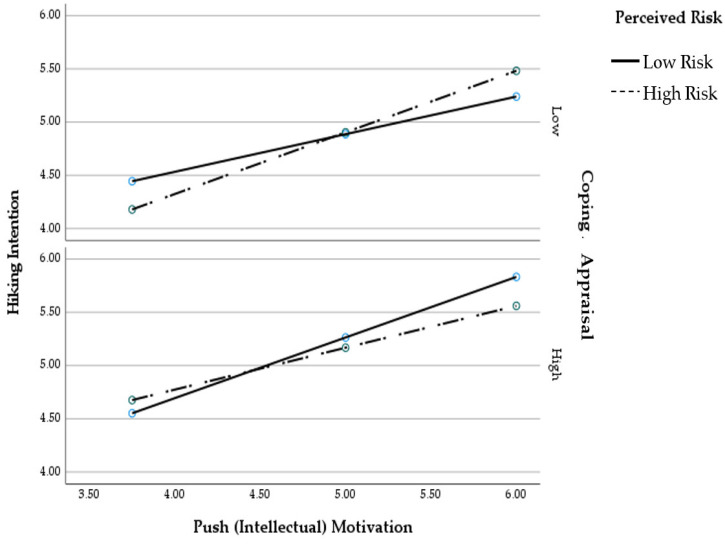
The three-way interaction of intellectual motivation and hiking intention.

**Figure 3 ijerph-19-04612-f003:**
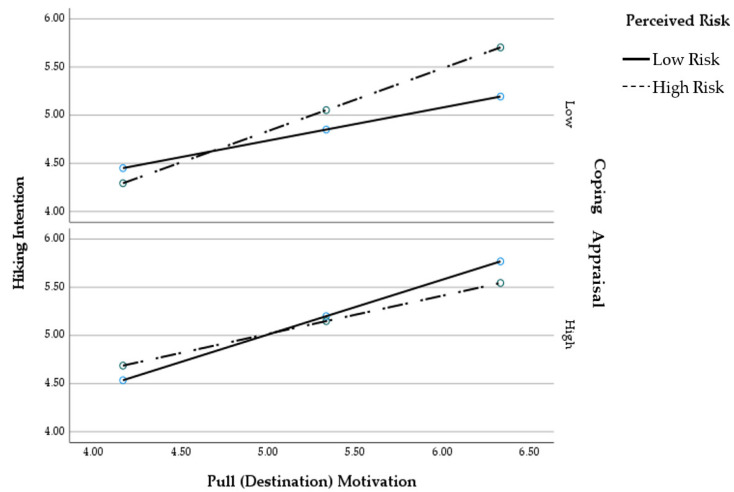
The three-way interaction of destination motivation and hiking intention.

**Table 1 ijerph-19-04612-t001:** Sociodemographic Characteristics of Participants (*N* = 407).

Category	Frequency	Percentage	Category	Frequency	Percentage
	* **N** *	%		* **N** *	%
**Gender**			**Education**		
Male	118	29.0	Junior high school and below	6	1.5
Female	289	71.0	High school/technical school	65	16.0
**Job**			College degree	122	30.0
Student	297	73.0	Bachelor degree	206	50.6
Government/Civil servant	5	1.2	Master degree and above	8	2.0
Enterprise managers	4	1.0	**Monthly Income**		
General Staff	23	5.7	None	28	6.4
Professional staff	15	3.7	Below 1500 CNY	198	31.3
Ordinary workers	12	2.9	1501–3000 CNY	100	24.6
Business service workers	5	1.2	3001–5000 CNY	35	8.6
Self-employed/contractors	3	0.7	5001–8000 CNY	28	6.9
Freelancer	22	5.4	Above 8000 CNY	15	3.7
Agriculture, forestry, animal husbandry, and fishery workers	3	0.7	Inconvenient to disclose	2	1.2
None	13	3.2	**Hiking frequency**		
Other	5	1.2	Less than once a month	228	56.0
**Hiking experience in one year**			Once a month	81	19.9
1–2 times	205	50.4	Twice a month	37	9.1
3–5 times	100	24.6	Above three times	61	15.0
5–10 times	52	12.8			
Over ten times	50	12.3			

**Table 2 ijerph-19-04612-t002:** Factor Loading, AVE and composite reliability (*N* = 407).

Scale	λ	AVE	CR
**Intellectual**			
To explore new ideas To learn about myself	0.62 0.75	0.54	0.83
To be creative	0.77		
To expand my knowledge.	0.79		
**Social**			
To meet new people	0.74	0.56	0.72
To build friendships and interact with others.	0.76		
**Mastery**			
To develop physical fitness To improve my skills and ability in hiking To challenge my abilities	0.70 0.69 0.76	0.48	0.73
**Avoidance**			
To refresh my mind	0.80	0.61	0.75
To relieve stress and tension	0.76		
**Destination**			
The hygiene and cleanliness of the hiking destination	0.75	0.46	0.84
The affordability of the hiking destination The accessibility of the hiking destination	0.67 0.62		
The well-organized hiking information system	0.71		
The cultural and historic resources of the hiking destination	0.66		
The wilderness and nature of the hiking destination	0.68		
**Perceived Risk (PR)**			
I am at risk of being a victim of COVID-19 while hiking.	0.63	0.53	0.77
Compared with other people, I easily infect with COVID-19 while hiking.	0.77		
The chance of someone my age of comparable physical condition getting COVID-19 is rather large.	0.78		
**Perceived Self-efficacy (SE)**			
I know how to prepare anti-COVID equipment and take coping measures effectively (e.g., masks) when I hike.	0.84	0.67	0.89
I consider getting the recommended vaccinations prior to go hiking.	0.85		
I can avoid trips with a large group of people or the crowded destination to reduce the risk of COVID-19 when I hike.	0.84		
I believe that I can reduce the risk of getting COVID-19 if I shorten the distance and time of the hiking trip.	0.73		
**Perceived Respond-efficacy (RE)**			
Preparing anti-covid equipment and coping measures	0.89	0.71	0.91
Obtaining recommended vaccinations	0.86		
Avoiding the trips with a large group of people	0.88		
Shortening the distance and time of hiking trip.	0.73		
**Hiking intention**			
How likely is it for you to go hiking in the current situation?	0.66	0.55	0.78
How much do you want to go hiking when you intend to do outdoor activities?	0.79		
How much are you willing to go hiking?	0.78		

Note: All factor loadings are significant at the *p* < 0.001. λ = factor loading, AVE= average variant extract, CR = Composite reliability.

**Table 3 ijerph-19-04612-t003:** Means, standard deviations and correlations among study variables.

Constructs	Mean	SD	1	2	3	4	5	6	7	8
1. Intellectual	4.95	1.16	**0.54**	0.63	0.61	0.44	0.51	0.06	0.08	0.32
2. Social	5.15	1.22	0.40	**0.56**	0.35	0.26	0.26	0.04	0.07	0.17
3. Mastery	5.12	1.14	0.78 **	0.59 **	**0.48**	0.46	0.44	0.05	0.07	0.36
4. Avoidance	5.64	1.10	0.66 **	0.51 **	0.68 **	**0.61**	0.51	0.02	0.16	0.25
5. Destination	5.28	1.03	0.72 **	0.51 **	0.66 **	0.71 **	**0.46**	0.03	0.13	0.27
6. Perceived risk	3.85	1.23	0.24 **	0.21 **	0.21 **	0.12 *	0.16 **	**0.53**	0.01	0.01
7. Coping appraisal	5.99	1.00	0.28 **	0.26 **	0.26 **	0.40 **	0.36 **	0.08	**0.69**	0.06
8. Hiking intention	5.04	1.01	0.57 **	0.41 **	0.60 **	0.50 **	0.52 **	0.09	0.25 **	**0.55**

Note: ** *p* < 0.01, * *p* < 0.05; The diagonal values written in bold are AVEs. Above the diagonal is the squared value of correlations.

**Table 4 ijerph-19-04612-t004:** The effects of the perceived risk and coping appraisal on the relationship between push motivations and hiking intention.

	*B*	*SE*	*t*	LL	UL	*p*
Constant	11.14	3.98	2.80	3.32	18.95	<0.01
**Intellectual** motivation (H1a)	−1.45	0.84	−1.72	−3.10	0.20	0.09
Perceived Risk	−2.34	1.13	−2.08	−4.55	−0.12	0.04
Coping appraisal	−1.35	0.62	−2.16	−2.57	−0.12	0.03
INT × PR (H2a)	0.49	0.23	2.13	0.04	0.94	0.03
INT × COP (H3a)	0.32	0.13	2.45	0.06	0.57	0.02
PR × COP	0.38	0.18	2.17	0.04	0.73	0.03
INT × PR × COP (H4a)	−0.08	0.04	−2.29	−0.15	−0.01	0.02
Education	0.02	0.05	0.30	−0.08	0.11	0.76
Gender	−0.33	0.09	−3.56	−0.51	−0.15	<0.01
*R*^2^ = 0.37, *MSE* = 0.67, *F* (9, 397) = 25.42, *p* < 0.001						
Constant	6.43	4.27	1.51	−1.95	14.82	0.13
**Social** motivation	−0.53	0.88	−0.61	−2.27	1.20	0.55
Perceived Risk	−1.05	1.17	−0.90	−3.35	1.24	0.37
Coping appraisal	−0.42	0.67	0.64	−1.73	0.89	0.53
SOC × PR	0.24	0.24	1.01	−0.23	0.70	0.31
SOC× COP	0.14	0.14	1.01	−0.13	0.40	0.32
PR × COP	0.17	0.18	0.95	−0.19	0.53	0.35
SOC × PR × COP	−0.04	0.04	−1.07	−0.11	0.03	0.28
Education	−0.02	0.06	−0.28	−0.13	0.09	0.78
Gender	−0.38	0.10	−3.68	−0.58	−0.18	<0.01
*R*^2^ = 0.22, *MSE* = 0.82, *F* (9, 397) = 12.34, *p* < 0.001						
Constant	6.37	4.54	1.40	−2.55	15.29	0.16
**Mastery** motivation	−0.54	0.92	−0.59	−2.36	1.27	0.56
Perceived Risk	−1.04	1.28	−0.81	−3.56	1.48	0.42
Coping appraisal	−0.48	0.70	−0.68	−1.85	0.90	0.50
MAS × PR	0.24	0.25	0.96	−0.26	0.74	0.34
MAS × COP	0.15	0.14	1.06	−0.13	0.43	0.29
PR × COP	0.13	0.20	0.65	−0.26	0.52	0.52
MAS × PR × COP	−0.03	0.04	−0.86	−0.11	0.04	0.39
Education	0.03	0.05	0.54	−0.07	0.12	0.59
Gender	−0.31	0.09	−3.51	−0.49	−0.14	<0.01
*R*^2^ = 0.40, *MSE* = 0.63, *F* (9, 397) = 28.94, *p* < 0.001						
Constant	6.65	4.78	1.39	−2.74	16.04	0.17
**Avoidance** motivation	−0.60	0.90	−0.67	−2.36	1.17	0.51
Perceived Risk	−0.73	1.35	−0.54	−3.39	1.93	0.59
Coping appraisal	−0.53	0.77	−0.70	−2.04	0.97	0.49
AVO × PR	0.19	0.25	0.77	−0.30	0.68	0.44
AVO × COP	0.15	0.14	1.07	−0.13	0.43	0.28
PR × COP	0.09	0.22	0.42	−0.33	0.52	0.67
AVO × PR × COP	−0.03	0.04	−0.67	−0.10	0.05	0.50
Education	<0.01	0.05	0.03	−0.10	0.11	0.97
Gender	−0.38	0.10	−3.85	−0.57	−0.18	<0.01
*R*^2^ = 0.29, *MSE* = 0.75, *F* (9, 397) = 17.70, *p* < 0.001						

Note: *B* = non-standardized regression coefficients, SE = standard error, LL = low limit, UL = upper limit, INT = intellectual motivation, SOC = social motivation, MAS = mastery motivation, AVO = avoidance motivation, PR = Perceived risk, COP = Coping appraisal.

**Table 5 ijerph-19-04612-t005:** The effects of the perceived risk and coping appraisal on the relationship between pull motivation and hiking intention.

	B	SE	*t*	LL	UL	*p*
Constant	12.96	4.96	2.61	3.20	22.79	0.01
Destination motivation (H1b)	−1.80	0.98	−1.82	−3.73	0.14	0.07
Perceived Risk	−2.90	1.42	−2.04	−5.70	−0.10	0.04
Coping appraisal	−1.65	0.78	−2.13	−3.18	−0.12	0.03
DES × PR (H2b)	0.61	0.28	2.18	0.06	1.15	0.03
DES × COP (H3b)	0.37	0.15	2.41	0.07	0.67	0.02
PR × COP	0.47	0.22	2.11	0.03	0.90	0.04
DES × PR × COP (H4b)	−0.10	0.04	−2.28	−0.18	−0.01	0.02
Education	<0.01	0.05	0.08	−0.10	0.11	0.94
Gender	−0.34	0.10	−3.52	−0.53	−0.15	<0.01
*R^2^ = 0.31, MSE = 0.73, F (9, 397) = 19.37, p < 0.001*						

Note: *B* = non-standardized regression coefficients, DES = Destination motivation, PR = Perceived risk, COP = Coping appraisal, Covariates: education and gender.

## Data Availability

Data are available on request due to privacy and ethical restrictions. The data presented in this study are available on request from the corresponding author. The data are not publicly available due to information privacy reasons.

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
