# Peer review of "Influence of COVID-19 Crisis on Motivation and Hiking Intention of Gen Z in China: Perceived Risk and Coping Appraisal as Moderators"

_ijerph, 2022, doi:10.3390/ijerph19084612_

Round 1
Reviewer 1 Report
The writing of this study is very good. However, the biggest flaw, which I think is fatal, is its sample. Even though it would be very difficult to get a random sample of 18-to-24 year-olds, other measures should have been taken to make the sample more representative of this young population. However, the authors did not make such attempts, and ended up with 29% male and 71% female. There is no information about whether roughly 73% of this age group (in the national population) has a college education. Furthermore, no demographic variables were treated as covariates (controls) in the analysis. As a result, no meaningful generalizations can be made based on its data.
The number two major problem, in my opinion, is the fact that the authors drew quite a few studies from sports/tourism/hospitality/marketing. In the context of the COVID-19 pandemic, the authors should have cited more studies in health communication instead. Both the concept and context of threat or risk in making a purchase are very different from a deadly virus.
The efforts are not wasted. The authors can see this a pilot study, then collect a better sample before writing a manuscript. The outcome can be promising.
Below are other flaws/concerns.
Page 1, first paragraph in the main text (lines 32-44), there is no reference [4]. I guess the [5] there should be [4] instead.
Page 2, line 48, I think there is a grammatical error (“the day hiking has received keep attention” Line 52, an “a” is missing between “in” and ‘relatively.”
Between lines 65 and 120, the author should do a better job explaining “internal,” “external,” “push” and “pull.” What was written (even with examples) is not sufficient.
Page 3, line 105 and line 113, [14] should be before [31] and [37].
Page 6, line 268, I think Cook’s value should be [63].
Page 7, first paragraph, please refer to Table 1. Second section (2.30), no need to talk about PROCESS in different places. Once is sufficient. In 3.1, line 327, the term “Composite reliability” is different from what’s in Table 1 (in which you use “construct reliability”).
Page 9, Table 2, if they are correlation coefficients, are all the numbers one off? For example, why is the correlation coefficient of #1 versus #1 .54? How does the same variable correlate with itself? Also, what does the subscript “a” mean? The latter is not in explained in the table.
Between lines 359 and 366, don’t we usually also report t-statistics and p-values?
Page 10, lines 407-408, if p = .05, it is not significant.
Page 11, line 431 is not a complete sentence.
Page 12, line 441, why is 4.13 the cut-off point? Also, again, if p =.05, it is not significant.
A few references don’t seem to be complete, such as [41] and [76].
Finally, in terms of writing, using the past tense to describe what other authors did (such as “discovered” or “found”) is correct. However, the knowledge that was generated should use the present tense.
Author Response
COMMENTS AND ACTIONS TAKEN
Influence of COVID-19 crisis on motivation and hiking intention of Generation Z in China: perceived risk and coping appraisal as moderators
First, we thank you for your careful review and helpful comments. We revised the paper accordingly. We feel the manuscript is strengthened and hope we satisfactorily addressed your concerns and that of the reviewers. As per the commentary and suggestions provided, we specifically discussed and addressed each. The reviewers’ comments and our reply to each follows.
Reviewer #1’s Comments and Author Responses
|
# |
Reviewer #1’s Comment |
Author Reply |
|
1 |
The writing of this study is very good. The biggest flaw is its sample. Even though it would be very difficult to get a random sample of 18-to-24 year-olds, other measures should have been taken to make the sample more representative of this young population. The authors ended up with 29% male and 71% female.
There is no information about whether roughly 73% of this age group (in the national population) has a college education. Furthermore, no demographic variables were treated as covariates (controls) in the analysis. As a result, no meaningful generalizations can be made based on its data.
|
Thank you for your positive feedback. Regarding our sample, we did our best to make the research population and sampling frame as similar as possible by using stratified sampling methods based on gender and geographic locations. However, we found out that male respondents on the survey panel we used were not active to answer the survey. Also, there were many missing cases for male respondents. These two reasons resulted in the gender unbalance. We believe what we can do at this moment is to report this gender unbalance issue in the limitation section.
We asked participants’ education background, which was missing in Table 1. We think that’s the origin of this confusion. Sorry for the inconvenience. Please see Table 1. Also, we re-run the data analysis. Based on our t-tests, we found no differences in terms of motivations between male and female participants (intellectual motivation; Mmale = 5.06, SDmale = 1.17, Mfemale = 4.90, SDfemale = 1.16, p = .23; social motivation; Mmale = 5.17, SDmale = 1.35, Mfemale = 5.14, SDfemale = 1.17, p = .84; masterly motivation; Mmale = 5.25, SDmale = 1.12, Mfemale = 5.07, SDfemale = 1.14, p = .16; avoidance motivation; Mmale = 5.64, SDmale = 1.06, Mfemale = 5.64, SDfemale = 1.12, p = .96; destination motivation; Mmale = 5.35, SDmale = 1.04, Mfemale = 5.26, SDfemale = 1.02, p = .42). Based on the results, we did not include gender as a covariate because it can potentially inflate the possibility of type 2 error. Still, we understand that this research is limited in its ability to generalize the findings. We reported this issue in the limitation section. Thank you. |
|
2 |
The number two major problem, in my opinion, is the fact that the authors drew quite a few studies from sports/tourism/hospitality/marketing. In the context of the COVID-19 pandemic, the authors should have cited more studies in health communication instead. Both the concept and context of threat or risk in making a purchase are very different from a deadly virus.
|
Thanks for the suggestion. We understand your concern and actually agree that threats/risk perception in leisure contexts are different than that in COVID-19 situations. We added some examples from health communication research on page 4 the paragraph starting from “People worry, feel …….” Thank you. |
|
3 |
The efforts are not wasted. The authors can see this a pilot study, then collect a better sample before writing a manuscript. The outcome can be promising. |
Although we understand your concern, we still believe that this piece can add knowledge to the literature as far as the limitations above are reported in the later section. Thank you. |
|
4 |
Page 1, first paragraph in the main text (lines 32-44), there is no reference [4]. I guess the [5] there should be [4] instead.
|
Thanks. We found that [5] is [4]. And the reference order is wrong so we changed both line (44) and line (660-663). |
|
5 |
Page 2, line 48, I think there is a grammatical error (“the day hiking has received keep attention” Line 52, an “a” is missing between “in” and ‘relatively.”
|
We revised them in line 48 (receive constant attention) and added “a” in line 52. |
|
6 |
Between lines 65 and 120, the author should do a better job explaining “internal,” “external,” “push” and “pull.” What was written (even with examples) is not sufficient.
|
Revised. Please see the 1.1. Push-pull motivations and hiking intention section. |
|
7 |
Page 3, line 105 and line 113, [14] should be before [31] and [37].
|
Revised. |
|
8 |
Page 6, line 268, I think Cook’s value should be [63].
|
We made an error in the reference list. The in-text citation was correct. Please see the revised reference list. |
|
9 |
Page 7, first paragraph, please refer to Table 1.
Second section (2.3), no need to talk about PROCESS in different places. Once is sufficient.
In 3.1, line 327, the term “Composite reliability” is different from what’s in Table 1 (in which you use “construct reliability”). |
Would you please specify your suggestion regarding Table 1?
For the section 2.3, we revised based on your suggestion.
Sorry for the confusion. We changed the title of Table 1. The correct term was “composite reliability.” |
|
10 |
Page 9, Table 2, if they are correlation coefficients, are all the numbers one off? For example, why is the correlation coefficient of #1 versus #1 .54? How does the same variable correlate with itself? Also, what does the subscript “a” mean? The latter is not in explained in the table. |
Please see the footnote underneath of the table. The diagonal values written in bold represent AVEs. Thank you for this. |
|
11 |
Between lines 359 and 366, don’t we usually also report t-statistics and p-values?
|
We believe it’s somewhat disciplinary differences though, reporting CIs is considered sufficient. However, we revised this section based on your suggestion. Please see the paragraph starting from Hypothesis 1a…… on page 9. |
|
12 |
Page 10, lines 407-408, if p = .05, it is not significant.
|
We believe this is correct. These values represent the Johnson-Neyman thresholds when using pick-a-point approach. We are showing the range of significance, and we made sure to report “lower than” or “higher than” the values (e.g., 3.25, 6.87). |
|
13 |
Page 11, line 431 is not a complete sentence.
|
Revised. Thank you. |
|
14 |
Page 12, line 441, why is 4.13 the cut-off point? Also, again, if p =.05, it is not significant.
|
Johnson-Neyman demonstrates the range of moderation. In this case, the statistics show that 4.13 is the threshold, which means that the moderation is significant when the value of moderator is lower than 4.13. We also made sure to report “lower than” the value (i.e., 4.13). |
|
15 |
A few references don’t seem to be complete, such as [41] and [76].
|
Revised. Thank you. We also check the whole reference list to make sure there is no errors.
|
|
16 |
Finally, in terms of writing, using the past tense to describe what other authors did (such as “discovered” or “found”) is correct. However, the knowledge that was generated should use the present tense.
|
Thanks for your suggestion. We revised the manuscript especially in the discussion section. We believe that the quality of the manuscript has improved much thanks to your help. |
Reviewer 2 Report
This is an interesting paper based on empirical research regarding some future trends in leisure activities, in this case hiking. The research seems to be well designed, the title reflects the content and its perspective (Chinese population).
Some suggestions:
- I would recommend to add some literature on the psychological effect of hiking for the personhood, i.e. FRANCK, Juan F. 2021. The Person at the Core of Psychological Science. Scientia et Fides 9(2), 15–33. DOI 10.12775/SetF.2021.016 and regarding the "coping" could be insightful the recent article in religious perceptive SERYCZYNSKA, Berenika.2021. Religious capital as a central factor in coping with the Covid-19: clues from an international survey, European Journal of Science and Theology 17 (2), 67-81
- The "risk" topic seems to be correctly focused, but it requires also one or two references to the current literature.
- What is the relationship between former willingness to hiking (before pandemic) and present situation? This not the main point, but would be interesting to follow this aspect in next publications
Finally, I think the sample, although might be a little bigger than 407, but it is sufficient for the study proposed in the paper. I would recommend the publication with these two suggestions
Congratulations for the interesting research that can be useful for agent specialized in tourism.
Author Response
Thank you for carefully reviewing our work. We believe that the quality of the manuscript has been improved a great deal thanks to your suggestions. Please see attached.

Reviewer 3 Report
This is an interesting and informative article that sheds light on many aspects of people's lives during a pandemic. The authors presented an study revealing new facets of the impact of the pandemic on different sides of generation Z lives. However, some minor and major issues should be addressed.
Point 1: What percentage of the entire population of China are zoomers? I would like to have an idea of the percentage of all major age groups (Zoomers, Millennials, Generation Alpha, Baby Boomers, etc.). Is this group a majority or a minority of the population?
Point 2: For what reason did the authors focus at the generation of zoomers? From the abstract and the introduction does not follow the answer to the question why this generation is so important and why it was taken for analysis.
Point 3: How important is this generation compared to the Millennials or the Alpha generation? In the Limitations section, the authors rightly point out that elderly should also be given attention. Why the authors ignored the Baby Boomer generation - the people who are most susceptible to COVID-19 - in their study?
Point 4: The first phrase of the abstract obviously raises no questions. However, there is no connection between it and the next phrase. The authors immediately proceed to the description of generation Z. After the first sentence, the authors should insert a sentence briefly listing all the main generations and only after that move on to a specific generation Z.
Point 5: What do the authors mean by "Chinese Generation Z between 18 to 24 in China"? did they weed out non-Chinese youth? Did they take into account nationality and citizenship?
Point 6: Table 1. The "Scale" column contains unnecessary abbreviations that are not further used in the table. Please, remove unused abbreviations.
Point 7: Table 1. The title of the table does not match its content. Please, make the title more descriptive, detailed and described the contents of Table 1.
Point 8: Table 2. “Note: The diagonal with a is the value of AVE.” it looks like some words are missing. In addition, where is this diagonal and what do the numbers from 1 to 8 mean? Please, make an additional line with the general designation of columns 1-8.
Point 9: Table 3. Please, fill out the empty cell. Make an additional line with the general designation of columns B, SE, t, LL etc.
Point 10: Figure 2. Please, give a general title for the both panels (“a” and “b”) of the drawing.
Point 11: Tables 4-7 are very similar in arrangement. Please consider whether you can combine them all or some of them into one.
Point 12: Figure 3. Please, give a general title for the both panels (“a” and “b”) of the drawing.
Author Response

(The authors gave the same response as above.)

Round 2
Reviewer 1 Report
The problem with using a convenience sample has not been addressed, such as comparing the demographics against the national population, or using demographics as control variables. Consequently, findings cannot be generalized.
Author Response
Thank you. We used demographic information as covariates – education and gender. We revised the tables and results accordingly.